# Effective Modulation by Lacosamide on Cumulative Inhibition of *I*_Na_ during High-Frequency Stimulation and Recovery of *I*_Na_ Block during Conditioning Pulse Train

**DOI:** 10.3390/ijms231911966

**Published:** 2022-10-08

**Authors:** Po-Ming Wu, Yu-Ching Lin, Chi-Wu Chiang, Hsin-Yen Cho, Tzu-Hsien Chuang, Meng-Cheng Yu, Sheng-Nan Wu, Yi-Fang Tu

**Affiliations:** 1Institute of Clinical Medicine, College of Medicine, National Cheng Kung University, Tainan 70101, Taiwan; 2Department of Pediatrics, National Cheng Kung University Hospital, College of Medicine, National Cheng Kung University, Tainan 70101, Taiwan; 3Department of Physical Medicine and Rehabilitation, National Cheng Kung University Hospital, College of Medicine, National Cheng Kung University, Tainan 70101, Taiwan; 4Institute of Molecular Medicine, College of Medicine, National Cheng Kung University, Tainan 70101, Taiwan; 5Department of Physiology, College of Medicine, National Cheng Kung University, Tainan 70101, Taiwan; 6Institute of Basic Medical Sciences, College of Medicine, National Cheng Kung University, Tainan 70101, Taiwan

**Keywords:** lacosamide (Vimpat^®^), voltage-gated Na^+^ current, transient (peak) Na^+^ current, late Na^+^ current, persistent Na^+^ current, window Na^+^ current, hysteresis, cumulative inhibition, current recovery

## Abstract

The effects of lacosamide (LCS, Vimpat^®^), an anti-convulsant and analgesic, on voltage-gated Na^+^ current (*I*_Na_) were investigated. LCS suppressed both the peak (transient, *I*_Na(T)_) and sustained (late, *I*_Na(L)_) components of *I*_Na_ with the IC_50_ values of 78 and 34 μM found in GH_3_ cells and of 112 and 26 μM in Neuro-2a cells, respectively. In GH3 cells, the voltage-dependent hysteresis of persistent *I*_Na_ (*I*_Na(P)_) during the triangular ramp pulse was strikingly attenuated, and the decaying time constant (τ) of *I*_Na(T)_ or *I*_Na(L)_ during a train of depolarizing pulses was further shortened by LCS. The recovery time course from the *I*_Na_ block elicited by the preceding conditioning train can be fitted by two exponential processes, while the single exponential increase in current recovery without a conditioning train was adequately fitted. The fast and slow τ’s of recovery from the *I*_Na_ block by the same conditioning protocol arose in the presence of LCS. In Neuro-2a cells, the strength of the instantaneous window *I*_Na_ (*I*_Na(W)_) during the rapid ramp pulse was reduced by LCS. This reduction could be reversed by tefluthrin. Moreover, LCS accelerated the inactivation time course of *I*_Na_ activated by pulse train stimulation, and veratridine reversed its decrease in the decaying τ value in current inactivation. The docking results predicted the capability of LCS binding to some amino-acid residues in sodium channels owing to the occurrence of hydrophobic contact. Overall, our findings unveiled that LCS can interact with the sodium channels to alter the magnitude, gating, voltage-dependent hysteresis behavior, and use dependence of *I*_Na_ in excitable cells.

## 1. Introduction

LCS, a functionalized amino acid, is an antiepileptic and analgesic drug available orally and intravenously in clinical practice [1,2,3,4,5,6]. It is commonly administrated in patients with epilepsy and occasionally in patients with trigeminal neuralgia or other neuropathic pain [1,5,7,8,9,10,11,12]. A growing body of evidence demonstrated the effectiveness and tolerability of LCS in patients with epilepsy [13,14,15,16,17,18,19]. Earlier reports have shown that this compound can decrease the frequency of interictal spikes as well as high-frequency oscillations in mesial temporal lobe epilepsy or refractory focal epilepsy [13,20]. The antiepileptic ability of LCS was proposed mainly because it selectively enhances the voltage-dependence of the slow inactivation of Na^+^ current (*I*_Na_) [5,7,21,22,23]. In addition to its antiepileptic effects, additional broad effects of LCS have been observed clinically [3,4]. For instance, a recent report showed that LCS could alter electrocardiographic changes in a status epilepticus animal model [24]. Another report showed that LCS could induce personality changes, which disappeared after its discontinuation [25]. Additional electrophysiological actions of LCS might explain its multiple therapeutic effects and need to be thoroughly established.

The voltage-gated Na^+^ currents form voltage-gated sodium channels (Na_V_) are critical for the generation and propagation of action potentials in excitable membranes [26,27]. These channels can briefly shift from the resting to the open state after depolarization, thereby allowing the flow of Na^+^ from the extracellular solution into the cell under the driving forces of the electrical and chemical gradients. After opening briefly in a voltage-dependent manner, the channels are shifted to the inactivated state(s), rendering *I*_Na_ intense but brief [27]. In addition to the voltage-dependence of the slow inactivation of *I*_Na_, LCS has also been shown to enhance the frequency-dependent inhibition of *I*_Na_ [28]. However, whether the accumulative inhibition of *I*_Na_ inactivation during repetitive depolarization and/or recovery from *I*_Na_ inactivation during the preceding conditioning pulse train could be perturbed by LCS has not been adequately investigated. In the current study, the following attempts were undertaken to evaluate how LCS could lead to any adjustments to the magnitude, gating kinetics, voltage-dependent hysteresis (Hys_(V)_), and use dependence of *I*_Na_ residing in two different excitable cells, pituitary tumor (GH_3_) cells and neuroblastoma (Neuro-2a) cells. The observations could help to delineate the delicate modulation of functional activities in excitable cells occurring in vivo.

## 2. Results

### 2.1. Effects of LCS on Voltage-Gated Na^+^ Current (I_Na_)

For the first stage of experiments, whether LCS would exert any perturbations on I_Na_ was tested. Pituitary tumor (GH3) cells were placed in Ca^2+^-free, Tyrode’s solution, which contained 10 mM tetraethylammonium chloride (TEA) and 0.5 mM CdCl_2_ to avoid interference by other types of ionic currents, such as K^+^ and Ca^2+^ currents. Upon membrane depolarization from a holding potential of −100 mV to −10 mV for 30 milliseconds (msec), the transient *I*_Na_ (*I*_Na(T)_) was elicited, and it could be attenuated dose-dependently by LCS (Figure 1A). At one minute after adding 30 μM or 100 μM LCS, the *I*_Na(T)_ amplitude measured at the beginning of the depolarizing pulse was attenuated to 1012 ± 29 pA (*n* = 8, *p* < 0.05) or 783 ± 22 pA (*n* = 8, *p* < 0.05) from a control value of 1212 ± 32 pA (*n* = 8), respectively. When LCS was removed, the current amplitude returned to 1207 ± 31 pA (*n* = 8).

In addition, late *I*_Na_ (*I*_Na(L)_) was measured at the end of the depolarizing test pulse. The extent of LCS-mediated inhibition of *I*_Na(L)_ was higher than that at the start of the pulse (i.e., *I*_Na(T)_). For example, 100 μM LCS decreased the *I*_Na(L)_ amplitude from 74 ± 9 pA to 29 ± pA (*n* = 8, *p* < 0.05). Meanwhile, upon exposure to 100 μM LCS, the slow component in the inactivation time constant (τ) of *I*_Na(T)_ decreased from 12.2 ± 2.1 to 8.1 ± 1.3 msec (*n* = 8, *p* < 0.05). The relationship between LCS concentration and the inhibition of *I*_Na(T)_ and *I*_Na(L)_ is constructed in Figure 1B. It demonstrated a differential dose-dependent LCS-mediated inhibition in *I*_Na(T)_ and *I*_Na(L)_ elicited by the rapid membrane depolarization. Based on a modified Hill equation described in Materials and Methods, the IC_50_ values required for exerting a suppressive effect on *I*_Na(T)_ and *I*_Na(L)_ were further estimated as 78 and 34 μM, respectively, reflecting that these two values are distinguishable.

The inhibitory effects of LCS on *I*_Na_ were verified in another kind of excitable cell: Neuro-2a cells. LCS (30 or 100 μM) also suppressed both the amplitude of *I*_Na(T)_ and *I*_Na(L)_ elicited by the rapid membrane depolarization (Figure 2A). Figure 2B shows a differential dose-dependent response of LCS-induced suppression of *I*_Na(T)_ and *I*_Na(L)_ in Neuro-2a cells. Based on a modified Hill equation, the IC_50_ values to inhibit the *I*_Na(T)_ and *I*_Na(L)_ amplitude were 112 and 26 μM, respectively. Thus, LCS can suppress the magnitude of *I*_Na(T)_ and *I*_Na(L)_ in a time- and concentration-dependent manner both in GH_3_ cells and Neuro-2a cells.

### 2.2. Effects of LCS on Persistent Na^+^ Current (I_Na(P)_) Triggered by Isosceles-Triangular Ramp Voltage (V_ramp_)

The persistent Na^+^ current (*I*_Na(P)_) is activated in the subthreshold voltage range and is also important in epilepsy by enhancing the repetitive firing capability of neurons. In this experiment, *I*_Na(P)_ was elicited in GH_3_ cells by an isosceles-triangular ramp voltage (V_ramp_), which consisted of an upsloping (ascending) voltage from −100 mV to +50 mV followed by a downsloping (descending) voltage from +50 mV back to −100 mV in 1 s [29,30,31]. Voltage-dependent hysteresis (Hys_(V)_) was observed in the instantaneous current–voltage (I–V) relationship of *I*_Na(P)_ and presented as two distinct Hys(V) loops of the *I*_Na_ amplitude: a high- (a counterclockwise direction) threshold loop and a low- (a clockwise direction) threshold loop (Figure 3A). Figure 3B shows the time course of the inhibitory effect of LCS (30 or 100 μM) on *I*_Na(P)_ amplitudes activated by double V_ramp_. Compared with the control situation without LCS, LCS reduced the *I*_Na(P)_ amplitudes on the loop of Hys_(V)_ (Figure 3B,C). The 30 and 100 μM LCS attenuated the *I*_Na(P)_ amplitudes at the level of −5 mV of the ascending limb to 245 ± 21 pA (*n* = 8, *p* < 0.05) and 212 ± 17 pA (*n* = 8, *p* < 0.05) from control values of 313 ± 24 pA (*n* = 8), and the *I*_Na(P)_ amplitudes at the level of −60 mV of the descending limb to 151 ± 14 pA (*n* = 8, *p* < 0.5) and 101 ± 11 pA (*n* = 8, *p* < 0.05) from control values of 198 ± 17 pA (*n* = 8), respectively. The attenuation of *I*_Na(P)_ was reversed by application of tefluthrin (Tef, 10 μM), which is an activator of *I*_Na_ [30].

### 2.3. Effect of LCS on Window I_Na_ (I_Na(W)_) Elicited by a Short Ascending V_ramp_

The next question is if the magnitude of *I*_Na(W)_ in response to the rapid ascending V_ramp_ can be modified by LCS. The instantaneous *I*_Na(W)_ was evoked by an ascending V_ramp_ from −80 to +40 mV for 30 msec (i.e., a ramp speed of 4 mV/msec) [32,33]. As shown in Figure 4A, the amplitude of *I*_Na(W)_ was markedly reduced within one minute while Neuro-2a cells were exposed to LCS. LCS (100 or 300 μM) decreased the amplitude of *I*_Na(W)_ measured at the level of −10 mV from a control value of 602 ± 27 pA (*n* = 7) to 442 ± 21 pA (*n* = 7, *p* < 0.05) or 296 ± 18 pA (*n* = 7, *p* < 0.05), respectively. After washout of LCS, the current amplitude at −10 mV was returned to 594 ± 24 pA (*n* = 7). Figure 4B illustrates a summary graph demonstrating the changes in ∆area of V_ramp_-elicited *I*_Na(W)_ measured at the voltage between −40 and +40 mV. It showed that LCS effectively diminished the *I*_Na(W)_’s area and Tef (10 μM) could reverse the LCS-mediated decrease in V_ramp_-elicited *I*_Na(W)_’s area.

### 2.4. Effect of LCS on the Cumulative Inhibition of I_Na_ during a Train of Depolarizing Pulses

*I*_Na(T)_ inactivation was previously demonstrated to accumulate before being activated during repetitive short pulses, which consisted of repetitive depolarization from −80 mV to −10 mV for 1 s with 40 msec in each pulse at a rate of 20 Hz [32,34]. To see the effects of LCS on the cumulative inhibition of *I*_Na_, the *I*_Na(T)_ or *I*_Na(L)_ amplitude with or without treatment of LCS was simultaneously measured at the beginning or the end of each depolarizing pulse. Without LCS, the *I*_Na(T)_ or *I*_Na(L)_ inactivation evoked by a 1 s repetitive depolarization showed decaying τ values of 86 ± 9 or 234 ± 22 msec (*n* = 8), respectively (Figure 5). This indicates a pronounced time-dependent decay of *I*_Na(T)_ or *I*_Na(L),_ which can be fitted with a single-exponential process. Under 30 and 100 μM LCS, the exponential time course of *I*_Na(T)_ or *I*_Na(L)_ elicited by the same train of depolarizing pulses was shortened to 64 ± 7 msec (*n* = 8, P < 0.05) and 31 ± 6 msec (*n* = 8, *p* < 0.05), or to 143 ± 9 msec (*n* = 8, *p* < 0.05) and 104 ± 6 msec (*n* = 8, *p* < 0.05), respectively. Thus, that accumulative inactivation of the current can be strikingly enhanced in the cells upon LCS exposure apart from a decrease in *I*_Na(T)_ and *I*_Na(L)_ amplitude.

The effects of LCS on the cumulative inhibition of *I*_Na_ were also verified in Neuro-2a cells. Consistently, LCS (100 μM) diminished the *I*_Na(T)_ magnitude and decreased the decaying τ value of *I*_Na(T)_ during repetitive depolarizations from a control value of 79 ± 9 msec to 39 ± 5 msec (*n* = 7, *p* < 0.05) (Figure 6A,B). In addition, the LCS-mediated reduction in decaying τ value of *I*_Na(T)_ could be partially reversed by veratridine (Figure 6C). Veratridine was a sodium channel agonist and was recently reported to modify the gating of the human NaV1.7 channel [35].

### 2.5. Modification by LCS of the Recovery Process of I_Na(T)_ Inactivation following Conditioning Train of Depolarizing Stimuli

Earlier investigations have disclosed a unique type of recovery from *I*_Na(T)_ inactivation evoked by a train of the preceding conditioning depolarizing stimuli [34,36,37]. The preceding conditioning train was composed of twenty 40 msec pulses separated by 5 msec intervals at −80 mV for 1 s (Figure 7A, top). Following such a conditioning train, the *I*_Na_ was produced by a two-step voltage protocol to measure the recovery time course of the current. The two-step voltage protocol included the first step that consisted of a 30 msec pulse from −80 to −10 mV and a second step that consisted of pulses for a variable length of time in a geometric progression (common ratio = 2) from −80 to −10 mV. The relative amplitude of recovery for current inactivation during this protocol was determined by the second step.

In the control situation (i.e., without LCS), the recovery from *I*_Na(T)_ inactivation elicited by the preceding conditioning depolarizing stimuli was noticed to emerge in a biphasic manner including a rapid rising recovery phase followed by a late slow phase (Figure 7A). It indicates a train of stimuli enabling the cells to resist the recovery from *I*_Na(T)_ inactivation. The recovery time course of this biphasic-manner *I*_Na(T)_ inactivation was constructed in Figure 7B. The experimental data points were well fitted with a sum of two exponential functions, i.e., fast (τ_fast_) and slow time constant (τ_slow_), as elaborated in Materials and Methods. Upon exposure to LCS (30 or 100 μM), both τ_fast_ and τ_slow_ in the recovery time course of current inactivation was increased (*p* < 0.05) compared to the control situation (Table 1).

## 3. Discussion

The principal finding in this study was that LCS could suppress *I*_Na_ in a time-, concentration- and frequency-dependent manner identified in GH_3_ and Neuro-2 cells. LCS induced differential inhibitions of *I*_Na(T)_ and *I*_Na(L)_ activated by a short depolarizing pulse. It also suppressed the high- or low-threshold amplitude of *I*_Na(P)_ elicited by the upright isosceles-triangular V_ramp_ at either the upsloping or downsloping limb leading to a striking reduction in Hys_(V)_ strength of the current. The accumulative inhibition of *I*_Na(T)_ and *I*_Na(L)_ during a train of depolarizing pulses was substantially enhanced by LCS, and the values of τ_fast_ and τ_slow_ in the recovery time course during the preceding conditioning train of depolarizing pulses were increased. Taken together, LCS could modify the magnitude, gating properties, use-dependence, and Hys_(V)_ behaviors of *I*_Na_, leading to the inhibition of *I*_Na(T)_, *I*_Na(L)_, *I*_Na(P),_ and *I*_Na(W)_.

In the present investigations, the non-equilibrium voltage-dependent hysteresis (Hys_(V)_) of *I*_Na(P)_ was observed by an upright isosceles-triangular V_ramp_ with a duration of 1 s (Figure 3). This implies that the *I*_Na(P)_ magnitude is contingent on the pre-existing state(s) or conformation(s) of the Na_V_ channel. There are two types of triangular V_ramp_-elicited *I*_Na(P)_, that is, low-threshold (i.e., activated at a voltage range near the resting potential) elicited upon the downsloping end of the triangular V_ramp_, and high-threshold (i.e., activated at a voltage range near the maximal *I*_Na(T)_) elicited on the upsloping end of such V_ramp_ [28]. LCS attenuated both types of triangular V_ramp_-elicited *I*_Na(P)_ of the Hys_(V)_ behaviors and Tef could effectively reverse this LCS-induced reduction in Hys_(V)_’s strength in the current. In the literature, this Hys_(V)_ behavior of triangular V_ramp_-elicited *I*_Na(P)_ has been demonstrated to link to the magnitude of background Na^+^ currents closely, and *I*_Na(L)_ and *I*_Na(P)_ during an extended period of time are likely to share the same Na_V_ channels [29,31,34,38,39]. This result indicates that LCS could diminish the background Na^+^ currents conductance and reduce the subthreshold potential or depolarization drive in these excitable cells.

In addition, the time-dependent decline in *I*_Na(T)_ and *I*_Na(L)_ during a 20 Hz train of depolarizing voltage commands (i.e., 40 msec pulses applied from −80 to −10 mV at a rate of 20 Hz for 1 sec) was observed in an exponential fashion, as shown in Figure 5 and Figure 6. This accumulative inhibition is a use-dependent property of *I*_Na(T)_ and *I*_Na(L)_ during rapid repetitive stimuli or high-frequency firing [32,34,40,41,42,43]. LCS could enhance this accumulative inhibition through reducing not only the amplitude but also the τ value of *I*_Na(T)_ and *I*_Na(L)_. The LCS-mediated enhancement was only reversed partially by veratridine, a sodium channel agonist. This means that LCS would lead to a “loss-of-function” change of Na^+^ channels and keep the Na^+^ channels in inactivated states during repetitive depolarization to prevent excessive excitability. In addition to LCS, a recent study demonstrated that other sodium blockers, such as lidocaine, also had the use-dependent inhibition of corneal nerve activity [44]. It is consistent with the current investigations.

It is important to emphasize that the LCS increased the τ_fast_ and τ_slow_ of recovery from the *I*_Na(T)_ block elicited by the preceding conditioning train pulse and enlarged the value of A, as summarized in Table 1. This preceding conditioning train pulse consisted of rapid repetitive pulses, which mimics high-frequency firing on excitable cells and would cause a large fraction of Na_V_ channels to shift toward the slowly recovering pool. The value of A indicates the fraction of the slow recovering pool of Na_V_ channels. Taken together, LCS could cause a larger fraction of the Na_V_ channels in an inactivated state and a longer recovery from inactivation after rapid repetitive stimuli or high-frequency firing. These results are partly relevant to the previous study that showed slow inactivation enhancement by LCS [21].

In this work, we further investigated how the protein of Na_V_ could be delicately docked with LCS by using PyRx software (https://sourceforge.net/projects/pyrx/; accessed on 17 July 2022). The predicted binding sites with LCS are demonstrated in Figure 8. The LCS could form hydrophobic contacts with several residues, including Gly 76, Trp 77, Phe 80, Gln 122, Leu 125, and Leu 126. Phe 80 can interact with the *N*’-acetylamino acid *N*’-benzyl amide unit of LCS, Gly 76 and Trp 77 can dock to the linker that connects the two-aryl group, while Gln 122, Leu 125, and Leu 126 are noted to interact with the terminal aromatic ring. These three functional groups (i.e., N’-acetylamino acid *N*’-benzyl amide unit, the linker that connects the two aryl groups, and the terminal aromatic ring) in the molecule could be essential for its Na_V_-channel blocking activity [7]. However, as the 4′ position of the structural moiety in safinamide (α-aminoamide) does not affect the slow inactivation of Na_V_ channels, the interaction with Phe 80 appears to be unimportant. The detailed structure of this Na_V_ channel, which is a particularly good exemplar for hNa_V_ pharmacology, was shown in an earlier study [45]. These results indicate that LCS can favorably interact with the amino-acid residues of the Na_V_ channel with an estimated binding affinity of −5.0. Kcal/mol, which is adjacent to the transmembrane region (i.e., position: 123–148) or membrane segment (i.e., position: 79–95) of the channel. Consequently, when LCS reaches the Na_V_ channels on the membrane, the interactions may raise the structural or steric constraints, thereby resulting in a substantial decrease in Na_V_-channel openings.

In summary, LCS could suppress *I*_Na_ in a time-, concentration-, and frequency-dependent manner and modify the magnitude, gating properties, use-dependence, and Hys_(V)_ behaviors of *I*_Na_, leading to inhibiting *I*_Na(T)_, *I*_Na(L)_, *I*_Na(P),_ and *I*_Na(W)_. Consequently, LCS could reduce the subthreshold potential, enhance the accumulative inhibition during repetitive depolarization, and prolong the recovery from inactivation after repetitive depolarization in excitable cells.

## 4. Materials and Methods

### 4.1. Chemicals, Drugs, and Solutions Used in This Study

Lacosamide (LCS, Vimpat^®^, *R*-enantiomer of 2-acetamido-*N*-benzyl-3-methoxy- propionamide, 2,3-diaminomaleonitrile, C_13_H_18_N_2_O_3_), tefluthrin (Tef), tetraethylammonium chloride (TEA), tetrodotoxin (TTX), and veratridine were supplied by Sigma-Aldrich (Merck, Taipei, Taiwan). For cell preparations, we obtained culture media, fetal bovine or calf serum, horse serum, L-glutamine, and trypsin/EDTA from HyClone^TM^ (Thermo Fisher, Kaohsiung, Taiwan). All other chemicals used in this work (e.g., CsOH, CsCl, CdCl_2,_ and HEPES) were of laboratory grade and taken from standard sources. Double-distilled water deionized through a Milli-Q^®^ purification system (Merck, Tainan, Taiwan) was used in all experiments.

The ionic composition of normal Tyrode’s solution buffered by HEPES was as follows (in mM): NaCl 136.5, CaCl_2_ 1.8, KCl 5.4, MgCl_2_ 0.53, glucose 5.5, and HEPES-NaOH buffer (pH 7.4). During the measurements recoding K^+^ currents, a patch electrode was filled with a solution (in mM): K-aspartate 130, KCl 20, MgCl_2_ 1, KH_2_PO_4_ 1, Na_2_ATP 0.1, Na_2_GTP 0.1, EGTA 0.1, and HEPES-KOH buffer (pH 7.2). To measure different patterns of voltage-gated Na^+^ current, we substituted K^+^ ions in the internal solution for equimolar Cs^+^ ions, and the pH value in the solution was adjusted to 7.2 by adding CsOH. The pipette solution and culture media presently used were filtered with an Acrodisc^®^ syringe filter that contains a 0.2 mm Supor^®^ nylon membrane (#4612; Pall Corp.; Genechain, Kaohsiung, Taiwan).

### 4.2. Cell Preparations

Both the mouse neuroblastoma cell line, Neuro-2a (N2a, BCRC-60026), and the pituitary adenomatous cell line, GH_3_ (BCRC-60015), were acquired from the Bioresource Collection and Research Center (Hsinchu, Taiwan). GH_3_ cells were in Ham’s F medium supplemented with 2.5% (*v*/*v*) fetal calf serum, 15% (*v*/*v*) horse serum, and 2 mM L-glutamine, while Neuro-2a cells were in Dulbecco’s modified Eagle’s medium with 10% (*v*/*v*) fetal bovine serum. These cells were maintained at 5% CO_2_ in a 37 ℃ water-jacketed incubator. Growth medium was replaced twice a week, and cells were split into subcultures once a week. Subcultures were made with trypsinization (0.025% trypsin solution (HyClone^TM^) containing 0.01% sodium *N*, *N*-diethyldithiocarbamate and EDTA). Electrophysiological measurements were undertaken five or six days after cells were cultured up to 60–80% confluence [30].

### 4.3. Electrophysiological Measurements

During the few hours before the measurements, we harvested Neuro-2a or GH_3_ cells with 1% trypsin-EDTA solution, and a few drops of cell suspension were rapidly placed into a custom-built recording chamber fixed on the stage of an inverted DM-IL microscope (Leica; Major Instruments, Tainan, Taiwan). We then suspended cells at room temperature (20–25 °C) in normal Tyrode’s solution until cells attached to the chamber’s bottom before the recordings were made. The pipettes used were fabricated from Kimax-51 glass tubing with an 1.5–1.8 mm outer diameter (#34500; Kimble, Dogger, New Taipei City, Taiwan) with a vertical two-stage puller (PP-83; Narishige, Taiwan Instrument, Tainan, Taiwan). When filled with different internal solutions, the electrodes presently used for measurements had a tip resistance of 3–5 MΩ. Ionic currents were examined in the whole-cell configuration of a modified patch-clamp technique with the use of either an Axoclamp-2B (Molecular Devices, Sunnyvale, CA, USA) or an RK-400 amplifier (Bio-Logic, Claix, France), as described elsewhere [28]. GΩ-seals were achieved in an all-or-nothing fashion and resulted in a dramatic improvement in the signal-to-noise ratio. The liquid junction potentials, which occur when the compositions in bath solution and those of the pipette internal solution are different, became zeroed shortly before GΩ-seal formation was achieved, and the whole-cell data were then corrected as previously desribed [30].

### 4.4. Data Recordings

The signals were monitored and digitally captured and stored online at 10 kHz or more in an ASUS ExpertBook laptop computer (Yuan-Dai, Tainan, Taiwan). For efficient analog-to-digital (A/D) and digital-to-analog (D/A) conversion to proceed, a Digidata^®^-1440A digitizer connected with a laptop computer via a USB 2.0 port was delicately operated by pClamp 10.6 software run under Microsoft Windows 7 (Redmond, WA, USA). Amplified current signals were low-pass-filtered at 2 kHz with an FL-4 four-pole Bessel filter (Dagan, Minneapolis, MN, USA). The voltage-clamp protocols comprising various rectangular and ramp waveforms were specifically designed and were thereafter imposed on the tested cells through D/A conversion. As pulse-train stimulation was needed, we used a dual output pulse stimulator (Astro-Med Grass S88X; Grass, West Warwick, RI, USA).

### 4.5. Data Analyses

To assess the dose–response curve of LCS-mediated inhibition on the peak (transient, *I*_Na(T)_) and sustained (late, *I*_Na(L)_) components of depolarization-activated *I*_Na_ present in GH_3_ or Neuro-2a cells, *I*_Na_’s were evoked by a 30 msec depolarizing pulse to −10 mV from a holding potential of −100 mV (indicated in the top part of Figure 1A and Figure 2A), and current amplitudes obtained with or without the exposure to different LCS concentrations (3 μM–1 mM) were measured at the beginning (*I*_Na(T)_) and end (*I*_Na(L)_) of the depolarizing pulses. The concentration needed to suppress 50% of the current amplitude (i.e., IC_50_) was appropriately determined according to the three-parameter logistic model (i.e., a modified form of the sigmoidal Hill equation) by use of goodness-of-fit assessments:(1)Relative amplitude= [LCS]−nH×(1−a)([LCS]−nH+IC50−nH)+a

In this equation, [*LCS*] = the LCS concentration; *n_H_* = the Hill coefficient; *IC*_50_ = the concentration required for a 50% inhibition. Maximal inhibition (i.e., 1 − *a*) was also approximated in this formula.

By a two-step voltage protocol with varying interpulse intervals in a geometric progression (common ratio = 2), the recovery time course of *I*_Na(T)_ from the block activated in response to the 1 sec conditioning pulse train was constructed, and the results acquired with or without the addition of LCS to GH_3_ or Neuro-2a cells were thereafter drawn by plotting the relative *I*_Na(T)_ amplitude (normalized with respect to the steady-state amplitude activated at 0.1 Hz). A basic assumption of the analyses is that the recovery time course of the current established under these experimental conditions can be reliably described by an exponential function [34]. Because the recovery time course in GH_3_ cells exhibits a rapid rising recovery phase followed by a late slow phase, there should be at least two underlying exponential terms. Accordingly, the data points showing a recovery time course with or without the addition of LCS were fitted to the exponential function with the biexponential process, i.e.,
(2)y=A×(1−e−tτfast)+B×(1−e−tτslow)
where *y* is the relative amplitude of *I*_Na_ at time *t*, *A* or *B* is the relative amplitude of each exponential component, and *τ_fast_* or *τ_slow_* is the fast or slow time constant in the recovery of *I*_Na_ block, respectively.

### 4.6. Curve-Fitting Approximations and Statistical Analyses

Linear or nonlinear curve fitting to experimental datasets in this work was made with the interactive least-squares procedure by using various tools, such as Microsoft Excel^®^-embedded “Solver” (Microsoft, Redmond, WA, USA) and the OriginPro^®^ 2021 program (OriginLab^®^; Scientific Formosa, Kaohsiung, Taiwan). The averaged results are presented as the mean ± standard error of the mean (SEM) with the sizes of independent samples (*n*) indicating cell numbers from which data were taken. The Student’s *t*-test (paired or unpaired) between the two different groups was applied. When differences among more than two groups were encountered, we performed either analysis of variance (ANOVA)-1 or ANOVA-2 with or without repeated measures, followed by the post hoc Fisher’s least significant difference test. Statistical significance (indicated with *, **, or ^+^ in the figures) was determined at a *p* value of <0.05.

## Figures and Tables

**Figure 1 ijms-23-11966-f001:**
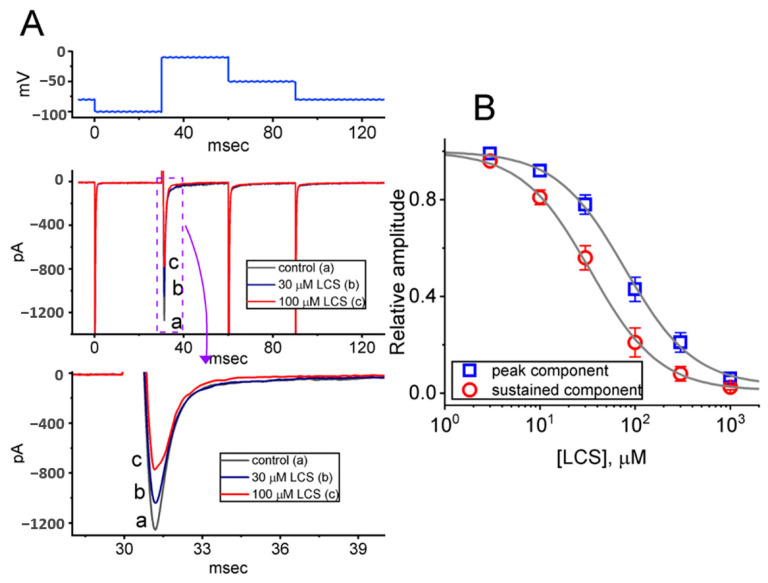
Effects of LCS on voltage-gated Na^+^ current (*I*_Na_) in GH_3_ cells. (**A**) The top part shows the voltage-clamp protocol. The middle part is the representative current traces obtained in (a, black) the control period (i.e., no LCS) and in 30 μM LCS (b, blue) or 100 μM LCS (c, red). The bottom part represents the expanded record from the purple dashed box. (**B**) Concentration-dependent relationship of LCS on transient (peak component, open blue squares) and late (sustained component, open red circles) *I*_Na_ evoked by short membrane depolarization (mean ± SEM; *n* = 8 for each point). The continuous gray line denotes the goodness-of-fit to a modified Hill equation, as stated in Section 4.

**Figure 2 ijms-23-11966-f002:**
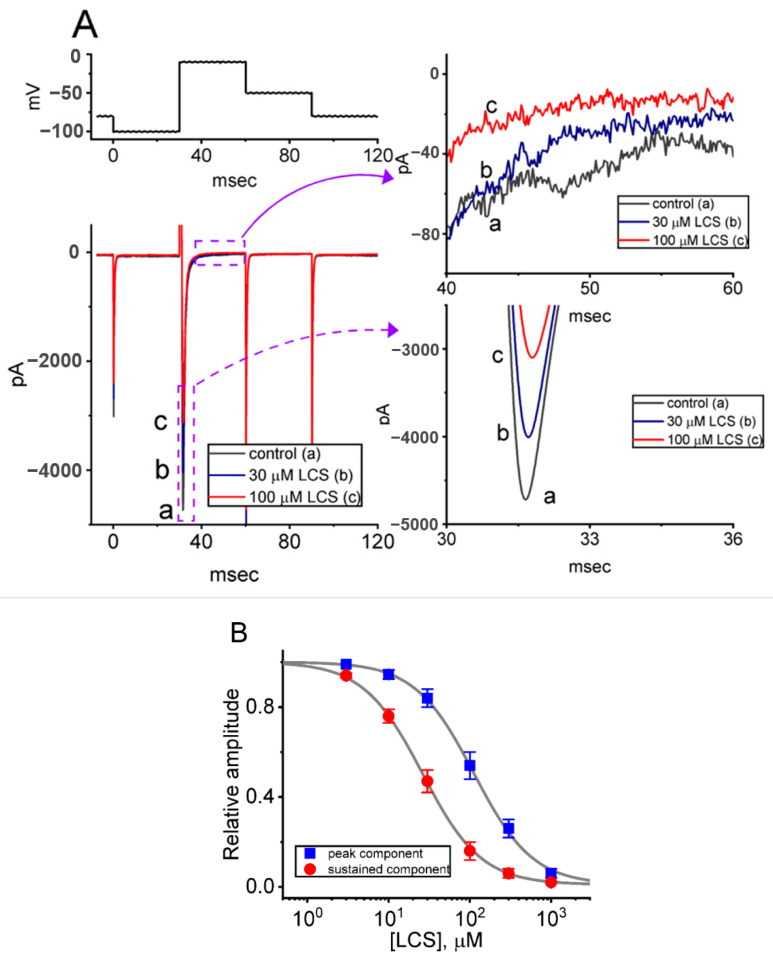
Effect of LCS on *I*_Na_ recorded from Neuro-2a cells. (**A**) The voltage-clamp protocol was on the top. The middle part was the representative current traces in response to depolarizing command pulses from −100 to −10 mV, a: control (i.e., no LCS, black); b: 30 μM LCS (blue); c: 100 μM LCS (red). The panels on the right side indicate the expanded records (i.e., *I*_Na(L)_ and *I*_Na(T)_) from each purple dashed box. (**B**) Dose-dependent relationship of LCS on *I*_Na(T)_ (peak component, blue squares) and *I*_Na(L)_ (sustained component, red circles) evoked by short membrane depolarization from −100 to −10 mV (mean ± SEM; *n* = 8 for each point). The continuous gray line represents the best fit to a modified Hill equation, as described in Section 4.

**Figure 3 ijms-23-11966-f003:**
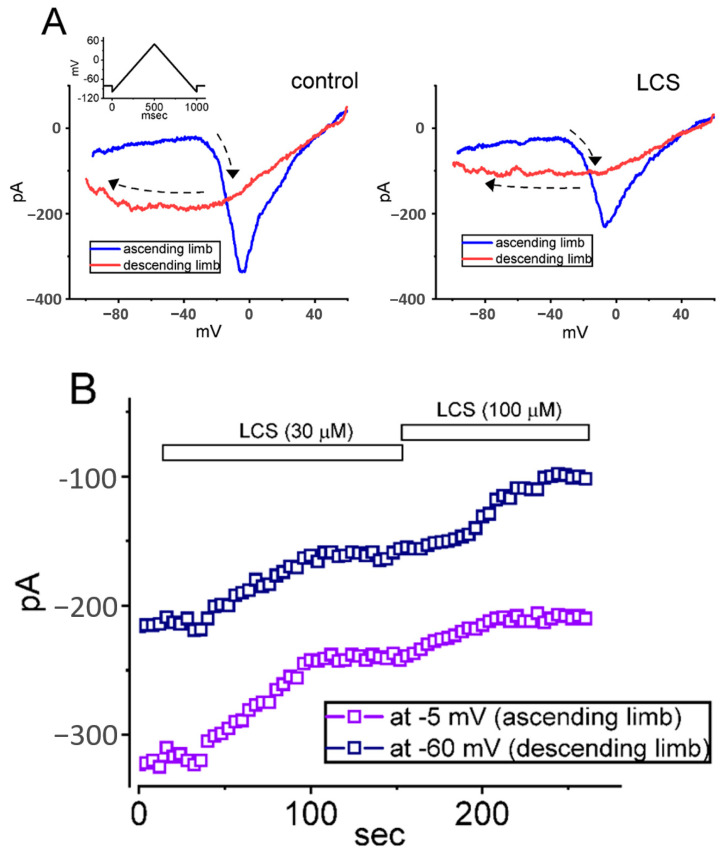
Inhibitory effect of LCS on persistent *I*_Na_ (*I*_Na(P)_) activated by an isosceles-triangular ramp voltage (V_ramp_) residing in GH_3_ cells. (**A**) The voltage-clamp protocol of V_ramp_ is shown on the top. The blue or red color indicates the current trajectory at the ascending or descending limb of Vramp, respectively. The dashed black arrow in each panel denotes the direction of the *I*_Na(P)_ trajectory by which the time goes during elicitation by such upright isosceles-triangular V_ramp_. (**B**) Time course of inhibitory effect of LCS (30 or 100 μM) during double V_ramp_. V_ramp_ was applied every 4 s, and *I*_Na(P)_ (open square) at −5 mV (ascending limb) or −60 mV (descending limb) was then measured. The horizontal bar shown above indicates the addition of LCS (30 or 100 μM). (**C**) Summary graphs demonstrating effects of LCS (30 or 100 μM) and LCS (100 μM) plus tefluthrin (Tef, 10 μM) on *I*_Na_ amplitude activated by the upsloping (left, high-threshold *I*_Na(P)_ (at −5 mV)) and downsloping (right, low-threshold *I*_Na(P)_ (at −60 mV)) V_ramp_ (mean ± SEM; *n* = 8 for each point). * Significantly different from controls (*p* < 0.05), ** significantly different from LCS (30 μM)-alone group (*p* < 0.05), and ^+^ significantly different from LCS (100 μM)-alone group (*p* < 0.05).

**Figure 4 ijms-23-11966-f004:**
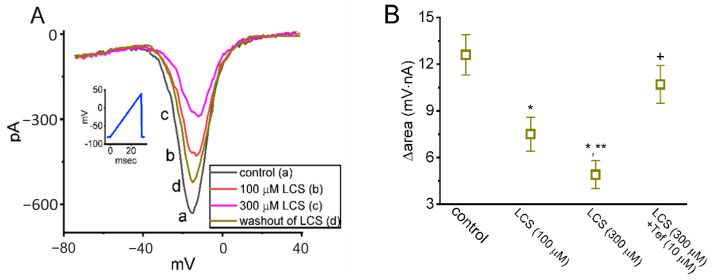
Modification by LCS of instantaneous window *I*_Na_ (*I*_Na(W)_) evoked in response to short ascending V_ramp_ in Neuro-2a cells. (**A**) Representative current traces were acquired in the control period (a, black), during cell exposure to 100 μM LCS (b, red) or 300 μM LCS (c, pink), and washout of LCS (d, brown). The voltage protocol is illustrated in the inset, and the downward deflection indicates instantaneous inward current (i.e., *I*_Na(W)_) elicited by a short ascending V_ramp_. (**B**) Summary graph demonstrating attenuating effect of LCS (100 or 300 μM) and LCS (300 μM) plus Tef (10 μM) on the ∆area of *I*_Na(W)_ (mean ± SEM; *n* = 7 for each point). Each area in this work was measured at the voltages ranging between −40 and +40 mV during the ascending V_ramp_. * Significantly different from control (*p* < 0.05), ** significantly different from LCS (100 μM) group (*p* < 0.05), and ^+^ significantly different from LCS (300 μM) group (*p* < 0.05).

**Figure 5 ijms-23-11966-f005:**
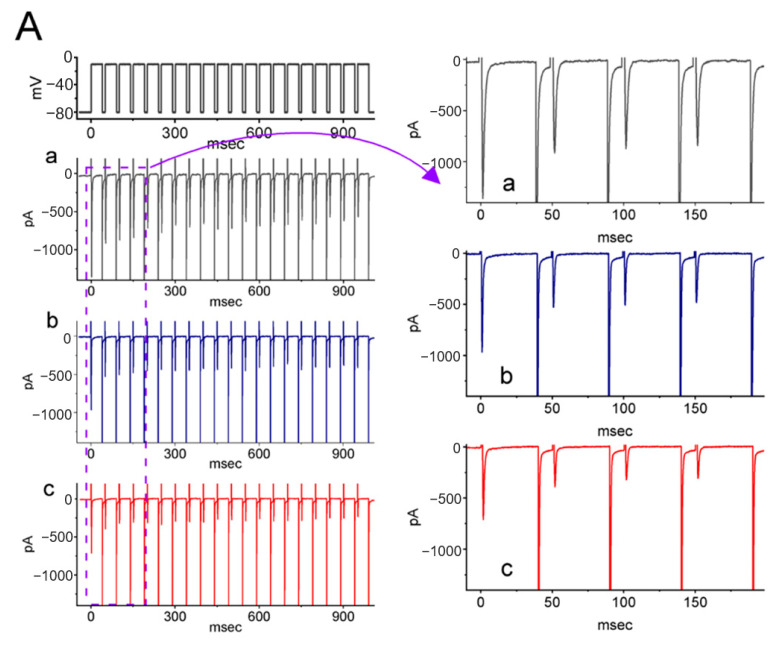
Effect of LCS on *I*_Na(T)_ activated by a train of depolarizing pulses in GH_3_ cells. The train consisted of twenty 40 msec pulses (stepped to −10 mV) separated by 10 msec intervals at −80 mV for 1 s. (**A**) Representative current traces were acquired in the control period ((a), no LCS, black) and during exposure to 30 μM ((b), blue) or 100 μM LCS ((c), red). The top part shows the voltage-clamp protocol applied. The right side of (**A**) represents the expanded records from the purple dashed box on the left side. (**B**) The relationship of the amplitude of *I*_Na(T)_ ((a), **(left)**) or *I*_Na(L)_ ((b), **(right)**) versus the pulse-train duration in the absence (black circles) and presence of 30 μM LCS (open blue circles) or 100 μM LCS (open red squares) was constructed (mean ± SEM; *n* = 8 for each point). The *I*_Na(T)_ or *I*_Na(L)_ amplitude was measured at the beginning or end of each depolarizing step, respectively.

**Figure 6 ijms-23-11966-f006:**
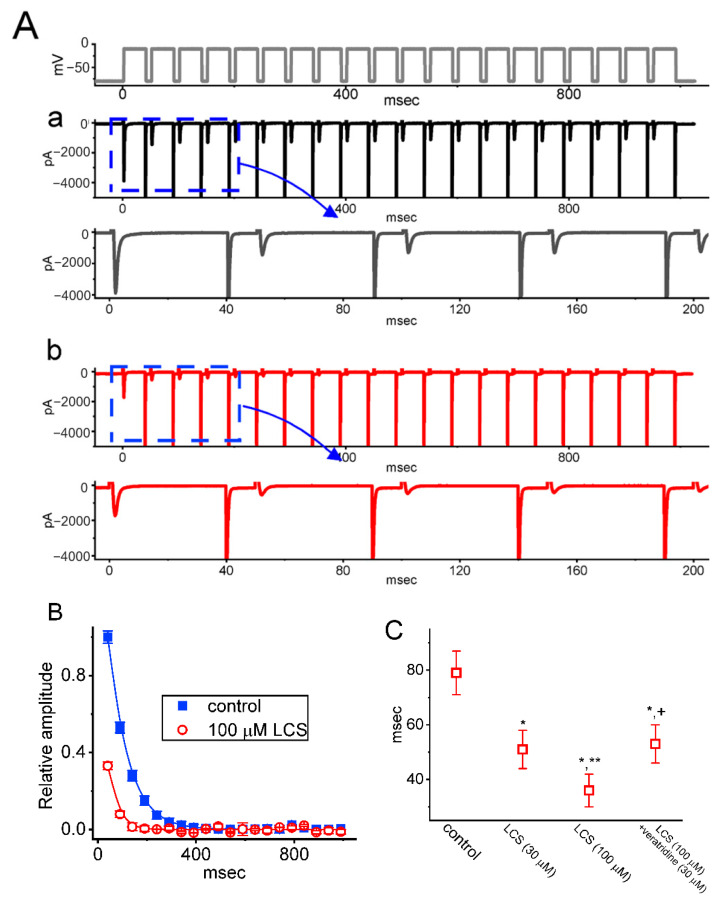
Effect of LCS on *I*_Na(T)_ during a train of depolarizing pulses in Neuro-2a cells. (**A**) The top part is the voltage-clamp protocol. Representative current traces during the control period ((a), black) or exposure to 100 μM LCS ((b), red) were presented. (**B**) The relationship of the relative amplitude of *I*_Na(T)_ versus the pulse duration with (open red circles) or without (blue squares) LCS (100 μM) (mean ± SEM; *n* = 7 for each point). Each smooth line is well fitted to a single exponential. (**C**) Summary graph demonstrating the effect of LCS (30 or 100 μM) and LCS (100 μM) plus veratridine (30 μM) on the τ value of *I*_Na(T)_ during repetitive depolarizing pulses (mean ± SEM; *n* = 7 for each point). * Significantly different from control (*p* < 0.05), ** significantly different from LCS (30 μM)-alone group (*p* < 0.05), and ^+^ significantly different from LCS (100 μM)-alone group (*p* < 0.05).

**Figure 7 ijms-23-11966-f007:**
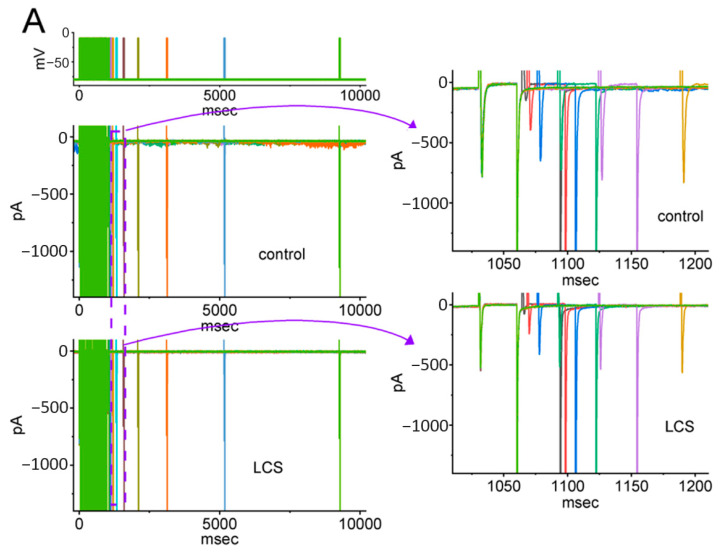
The recovery of *I*_Na_ inactivation during the train of conditioning depolarizing pulses modified by the presence of LCS. (**A**) The voltage-clamp protocol is shown on the top. Representative current traces were taken in the control period (middle part) and during exposure to 100 μM LCS (lower part). The right side of (**A**) denotes the expanded records from the purple dashed box on the left side. (**B**) The relationship of the relative amplitude of *I*_Na(T)_ versus the interpulse interval was obtained in the absence (blue circles) and presence (open red circles) of 100 μM LCS (mean ± SEM; *n* = 8 for each point). The blue or red smooth curve taken with or without the application of LCS was least-squares fitted with a two-exponential function, as detailed in Section 4, while the parameters are illustrated in Table 1.

**Figure 8 ijms-23-11966-f008:**
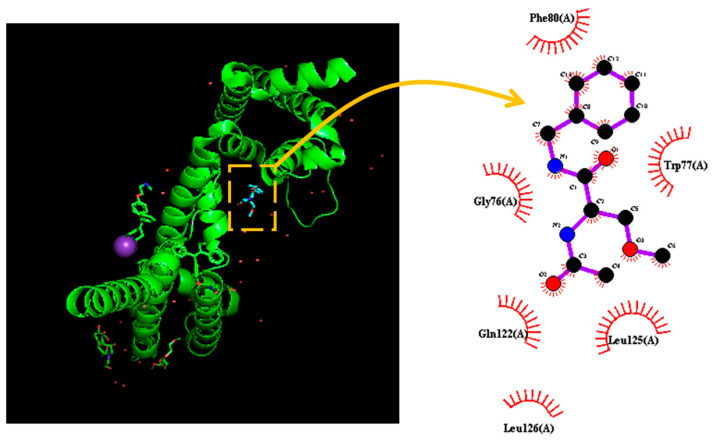
Predicted docking results demonstrating an interaction with Na_V_ channel and lacosamide (LCS). The protein structure of the Na_V_ (SCN) channel was acquired from PDB (PDB ID: 6Z8C), while the chemical structure of LCS was from PubChem (Compound CID: 219078). The structure of the Na_V_ channel was docked with the LCS molecule through PyRx (https://pyrx.sourceforge.io/, accessed on 5 June 2022). The diagram of the interaction between Na_V_ and the LCS molecule was generated by LigPlot^+^ (https://www.ebi.ac.uk/thornton-srv/software/LIGPLOT/, accessed on 15 June 2022). Note that the red arcs with spokes that radiate toward the ligand (i.e., LCS, in the center) represent the hydrophobic contacts.

**Table 1 ijms-23-11966-t001:** Summary of data demonstrating the parameter values for the modulatory effect of LCS on the recovery of *I*_Na_ block during the preceding train pulse observed in pituitary GH3 cells. These parameters are elaborated in detail in Section 4.

	N	τ_fast_ (msec)	τ_slow_ (msec)	A	B
Control	8	12.2 ± 0.4	885 ± 16	0.71 ± 0.04	0.28 ± 0.02
LCS (30 μM)	8	13.6 ± 0.6 *	972 ± 17 *	0.78 ± 0.04 *	0.22 ± 0.02 *
LCS (100 μM)	8	14.1 ± 0.6 *	1045 ± 19 *	0.81 ± 0.04 *	0.18 ± 0.02 *

All values are mean ± SEM. * Significantly different from controls (*p* < 0.05).

## Data Availability

The original data are available upon reasonable request to the corresponding author.

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
