# Peer review of "Effective Modulation by Lacosamide on Cumulative Inhibition of INa during High-Frequency Stimulation and Recovery of INa Block during Conditioning Pulse Train"

_ijms, 2022, doi:10.3390/ijms231911966_

Round 1

Reviewer 1 Report

In this work, authors show that lacosamide, a widely clinically used anti-convulsant and analgesic agent, acts directly on voltage-gated sodium channels, in a way that their activity get disminished. Although this interaction has been indirectly postulated by previous studies (Kovács et al 2016., 2016. Eur J Pharmacol), there is currently a lack of studies showing direct evidences of voltage-gated sodium channels inhibition by lacosamide. Overall, the manuscript is well-written and data support the conclusions. The data have been obtained by means of electrophysiological recordings on cell lines and Nav channel related pharmacology. That said, a few concerns arose during reviewing:

- Figure 1A and 2A. Protocol is not fully explained. I mean, the reason why to evoke capacitative currents before and after sodium currents is not indicated at any point in the draft, and this turns out to be confusing since they look pretty similar to Na+ currents dsiplayed at the second pulse.

- Figure 3. It would be clearer for a better understanding to show the time course of the sodium current upon the isosceles-triangular voltage ramp and not only the I-V relationship.

- Figure 4. Washout of LCS should be shown in  figure 4

Discussion. Line 292 to 299. This point of discussion could be expanded. Indeed, a recent paper show use-dependent inhibition of corneal nerve activity by other sodium blockers such as lidocaine (Luna et al 2021, IOVS)

Author Response

Reviewer one:

In this work, authors show that lacosamide, a widely clinically used anti-convulsant and analgesic agent, acts directly on voltage-gated sodium channels, in a way that their activity get disminished. Although this interaction has been indirectly postulated by previous studies (Kovács et al 2016., 2016. Eur J Pharmacol), there is currently a lack of studies showing direct evidences of voltage-gated sodium channels inhibition by lacosamide. Overall, the manuscript is well-written and data support the conclusions. The data have been obtained by means of electrophysiological recordings on cell lines and Nav channel related pharmacology. That said, a few concerns arose during reviewing:

- Figure 1A and 2A. Protocol is not fully explained. I mean, the reason why to evoke capacitative currents before and after sodium currents is not indicated at any point in the draft, and this turns out to be confusing since they look pretty similar to Na+ currents dsiplayed at the second pulse.

Ans:

Thanks for the reviewer’s comments.

The capacitative currents shown before and after voltage-gated sodium currents have been truncated in the present study. The INa traces in response to brief rapid depolarization were noticed to emerge in the ‘reverse’ direction (i.e., inward current in response to rapid depolarizing pulse), while capacitative current traces were expected to display in the ‘same’ direction (i.e., outward capacitative current activated by short depolarizing pulse, whereas inward capacitative current by rapid hyperpolarizing pulse). Hence, the current traces shown in Figures 1 and 2 are clearly illustrated. The magnified current traces are also shown in Figure 1A and 2A.

- Figure 3. It would be clearer for a better understanding to show the time course of the sodium current upon the isosceles-triangular voltage ramp and not only the I-V relationship.

Ans:

Thanks for the comments provided by the reviewer.

An additional panel (i.e., Figure 3B) regarding the time course of inhibitory effects of LCS on INa(P) activated by each isosceles-triangular ramp voltage was added in the revised manuscript. The text and the legend in Figure 3 was also accordingly revised (Lines 141-142, 158-161).

- Figure 4. Washout of LCS should be shown in  figure 4

Ans:

Thanks for the reviewer’s comment.

The current trace after washout of LCS was added in Figure 4A (d) in the revised manuscript.

Discussion. Line 292 to 299. This point of discussion could be expanded. Indeed, a recent paper show use-dependent inhibition of corneal nerve activity by other sodium blockers such as lidocaine (Luna et al 2021, IOVS)

Ans:

Thank you very much for pointing out this issue.

The findings of this article were added and discussed in the 3rd paragraph of the discussion section in this revised manuscript (reference 44, Line 319-321).

Author Response

Reviewer 2:

The article „Effective modulation by lacosamide on cumulative inhibition of INa during high-frequency stimulation and recovery of INa block during conditioning pulse train” presents an analysis of the blocking effect of lacosamide on the whole cell sodium currents in cancer lines of neuronal origin. Lacosamide is an anti-convulsant and analgesic drug used in epilepsy and neuropathic pain. The mode of action of lacosamide is important for understanding the therapeutic effect of lacosamide. To carry out the study the authors used different voltage clamp protocols in the patch-clamp technique. The research conducted by authors revealed that lacosamide reduces the transient and sustained/persistent sodium currents, accelerates the inactivation time course in a train of depolarizing pulses and decreases recovery time, all in concentration dependent manner. In Discussion authors predict binding site of LCS.

The topic addressed in the manuscript is presented clearly. Half of the cited publications are from the last five years. There is a number of self-citations, which in my opinion are not necessary in some places (i.e. line 82, 133, 161). Citation 46 (also self-citation) seams to be not relevant to the text for which it was used (line 292-294). Methodology and presentation of the results are correct and should be easy to reproduce in other laboratory. Figures and tables are appropriate and show the data clearly (though some modification would be useful, look at comment 5). Conclusion are supported by the data.

Ans:

Thanks for the comments of the reviewer. We deleted some unnecessary references (reference 29, 30, 36, 46) from the original citation list.

Specific comments:

  1. line 264 What does it mean that “LCS could suppress INa in a time-…….. and frequency-dependent manner in this research. Could you explain it more thoroughly?

Ans:

Thanks for the comments provided by the reviewer.

The current findings showed that LCS could result in a reduction in the slow component of the inactivation time constant of INa(T) (2nd paragraph of the results, Line 95), indicating that the inhibitory effects of LCS on INa(T) occur in a ‘time-dependent’ manner. The inhibitory effects of LCS on INa(T) activated during a train of depolarizing pulse were noted to occur in a frequency-dependent manner (Figure 5, Figure 6).

  1. line 278 In Fig 3 ramp has duration of 1 s, why there is 320 ms?

Ans:

Thank you very much for pointing out this error. It is 1 second. We amended it in this revised manuscript (Line 296-297)

  1. Are INaL) and INa(P) carried by the same Nav channels? Could you address it in Discussion?

Ans:

Yes, INa(L) and INa(P) during an extended period of time are likely to share the same NaV channels (Taddese et al. Neuron 2002;33(4): 587-600). We add this description in the 2nd paragraph of the discussion section (Line 305-306).

  1. How the presented results relate to slow inactivation enhancement reported for LCS action on INa (citation 21)?

Ans:

Thanks for the critical comments provided by the reviewer.

In this study, the recovery of INa inactivation during the train of conditioning depolarizing pulses was perturbed by LCS (Table 1 and Figure 7). The slow component in recovery time course after the conditioning pulse train was enhanced by adding LCS. These results were partly relevant to the previous study, which showed slow inactivation enhancement by LCS (Errington et al. Mol Pharmacol 2008;73(1): 157-69). We added these descriptions in the 4th paragraph of the discussion section (Line 330-332).

  1. It would be useful to remove the capacitive current peaks, as they reduce the clarity of the current diagrams.

Ans:

Thanks for the reviewer’s comments.

The capacitative currents in the present study have been truncated. The INa traces in response to brief rapid depolarization were noticed to emerge in the ‘reverse’ direction (i.e., inward current in response to rapid depolarizing pulse). Conversely, capacitative current traces were usually in the ‘same’ direction (i.e., outward capacitative current activated by short depolarizing pulse, while inward capacitative current by rapid hyperpolarizing pulse). Thus, the current diagrams shown in the present study should be acceptable.
